# Factors Influencing Walking and Exercise Adherence in Healthy Older Adults Using Monitoring and Interfacing Technology: Preliminary Evidence

**DOI:** 10.3390/ijerph17176142

**Published:** 2020-08-24

**Authors:** Andrea Albergoni, Florentina J. Hettinga, Wim Stut, Francesco Sartor

**Affiliations:** 1Department of Biomedical Sciences for Health, University of Milan, 20133 Milan, Italy; s4794406@studenti.unige.it; 2Centro Polifunzionale di Scienze Motorie, University of Genoa, 16132 Genoa, Italy; 3Department of Neuroscience (DINOGMI), University of Genoa, 16148 Genoa, Italy; 4Department of Patient Care and Measurements, Philips Research, 5656AE Eindhoven, The Netherlands; 5School of Sport Rehabilitation and Exercise Sciences, University of Essex, Colchester CO4 3WA, UK; florentina.hettinga@northumbria.ac.uk; 6Department of Sport, Exercise and Rehabilitation, Northumbria University, Newcastle NE1 8ST, UK; 7Department of Chronic Disease Management, Philips Research, 5656AE Eindhoven, The Netherlands; wim.stut@philips.com; 8College of Health and Behavioural Science, Bangor University, Bangor LL57 2EF, UK

**Keywords:** activity pacing, wearables, profiling

## Abstract

Background: Monitoring and interfacing technologies may increase physical activity (PA) program adherence in older adults, but they should account for aspects influencing older adults’ PA behavior. This study aimed at gathering preliminary wrist-based PA adherence data in free-living and relate these to the influencing factors. Methods: Ten healthy older adults (4 females, aged 70–78 years) provided health, fatigue, activity levels, attitude towards pacing, and self-efficacy information and performed a 6 min-walk test to assess their fitness. After a baseline week they followed a two-week walking and exercise intervention. Participants saw their progress via a purposely designed mobile application. Results: Walking and exercise adherence did not increase during the intervention (*p* = 0.38, *p* = 0.65). Self-efficacy decreased (*p* = 0.024). The baseline physical component of the Short Form Health Survey was the most predictive variable of walking adherence. Baseline perceived risk of over-activity and resting heart rate (HR_rest_) were the most predictive variables of exercise adherence. When the latter two were used to cluster participants according to their exercise adherence, the fitness gap between exercise-adherent and non-adherent increased after the intervention (*p* = 0.004). Conclusions: Risk of over-activity and HR_rest_ profiled short-term exercise adherence in older adults. If confirmed in a larger and longer study, these could personalize interventions aimed at increasing adherence.

## 1. Introduction

### 1.1. Background

Physical inactivity is one of the most important risk factors for mortality [1], and this is particularly true in older adults [1]. In turn, physical activity (PA) provides a number of health benefits. It reduces the risks of cardio-respiratory and metabolic diseases, osteoporosis, depression, and breast and colon cancer [1]. Furthermore, in older adults, PA helps to reduce the risk of falls by nearly 30% [1]. PA guidelines for Americans and the World Health Organization recommend 150 min a week of multicomponent physical activity training, or 75 min a week of vigorous-intensity aerobic PA, performed in bouts of at least 10 min duration or as much as their abilities and conditions allow [1,2]. Unfortunately, the notion that physical activity is somewhat unnecessary or even potentially harmful still exists amongst some individuals of the senior population [3]. Adherence to PA programs or guidelines is still low, especially in older adults, and it decreases with age [4,5,6,7] and health condition [8]. The main observed barriers to adherence to PA programs were identified as lack of social support, fear of injuries, stress, depression, and mental disorders [9,10]. Moreover, the negative perception of one’s own physical capacity could influence adherence [11]. In the older population, low levels of PA adherence are also associated with fatigue induced by chronic illnesses [12]. Conversely, facilitators of adherence in seniors are a good health state, lower body mass index, high social status, and higher education [7,10]. Self-efficacy appeared to be one of the strongest predictors for PA long-term levels [13]. Being highly adherent to a PA program positively influences physical function, performance, and pain relief [8]. A mobile application is able to combine elements, such as goal setting, self-monitoring and action planning known to help facilitate long term adherence to PA programs [14]. Furthermore, a mobile application can be a strong facilitator, as it can objectively monitor the actual behavior as it occurs and provide timely and highly personalized feedback.

### 1.2. Role of Technology

In our recent systematic review, we observed that technology is mainly used to monitor PA but is not yet consistently used to improve adherence to PA and exercise programs in patients with chronic diseases experiencing fatigue [15], and the same may be said for the older population. According to a recent systematic review, technology-based exercise interventions seem to promote good adherence amongst older adults [16]. However, a lack of details on adherence did not allow for meta-analysis of the adherence rate [16]. Therefore, our rationale is that the use of monitoring and user-interface technology can promote adherence to PA programs in older adults, provided that the feedback is clear, personalized and dynamic. We developed a mobile application with the specific aim of increasing PA program adherence. It was designed to allow users to plan their PA sessions, as prescribed by a health professional, and receive clear straightforward feedback at any moment [17]. The progress on the program could be reviewed by the health professional and updated according to the circumstances. The users were able to decide what day to perform the prescribed training. Training longer than prescribed was not rewarded by the app. The feedback was centered on the adherence to the prescribed program. Our Physical Activity Cardiorespiratory Exercise (PACE) mobile application was synchronized with a validated wrist-based activity monitor [18,19,20], which provided optical heart rate data, used for computing the exercise minutes and acceleration data from which walking minutes were derived. We hypothesized that a higher degree of program personalization could be achieved by understanding the app users’ physical behavior influencing factors.

### 1.3. Influencing Factors

In order to gather further insights in the factors which may influence older adults’ physical behavior and thus the adherence to the program provided by the PACE app, general health status, presence of fatigue, attitude towards activity pacing as done in Abonie et al. 2020 [21], and self-efficacy were monitored. General health state is known to considerably influence PA levels, especially in older adults [7]. It has been also shown that older adults can display a more rapid fatigability when compared to a younger population [22], and fatigue reduces intensity and frequency of PA sessions [23].

Next to health status and fatigability, the way that older adults perceive PA and their approach to it can play a major role in their adherence. Recently, the concept of “race pacing” (i.e., the strategy to distribute energy and manage fatigue during a long-distance race [24]) has been exploited in chronic disease and special populations [25,26]. In this new context, activity pacing is referred to as a coping strategy to manage fatigue consisting of optimal human energy management and activity splitting into more feasible exercise portions [27]. It also includes activity planning and goal setting to increase activity levels [28]. As explained by White et al. [29], activity pacing strategies serve to decrease fatigue or improve daily activity levels, especially in patients with chronic diseases.

The perception that older adults have of PA is also reflected in their self-efficacy. Self-efficacy, defined as “an individual’s belief in his or her capacity to execute behaviors necessary to produce specific performance attainments” [30], seems to be one of the most determinant factors to start and maintain a PA program [13]. Older adults tend to have a low self-efficacy when compared to middle-age people [31].

Finally, high cardiorespiratory fitness (CRF) is usually associated with higher habitual PA levels [32]. Therefore, it is important to evaluate the starting CRF level of older adults when enrolling them in a PA and exercise program.

Taking all these considerations together, we have set up a pilot study to examine the influence of each of these factors in determining short term adherence to our PACE mobile application.

### 1.4. Objectives

This pilot study had a number of objectives. The primary aim was to gather preliminary wrist-based activity and exercise program adherence data in free-living older adult individuals and to relate these to a number of influencing factors: (i) general health status, (ii) perceived fatigue, (iii) attitudes towards activity pacing, (iv) physical activity, and (v) self-efficacy reported by the older adults. Secondly, this pilot study was used to run a data driven adherence cluster analysis in order to identify influencing factors that would best predict who would adhere to the program, with the future aim in mind to tailor programs according to user’s profiles. Finally, this pilot study was used to test how the PACE mobile application was received by this age group, and whether the participants understood and appreciated the feedback on their adherence to the PA and exercise programs. It is helpful to point out that this pilot study was not designed to accept or reject the hypothesis that the PACE app would be able to increase adherence, as this would require an adequately sized randomized control trial.

## 2. Methods

### 2.1. Participants and Study Design

The study protocol was reviewed and approved by the Internal Committee of Biomedical Experiments of Philips Research, in conformity with the Declaration of Helsinki. Ten older adult participants (6 males) were recruited by means of a specialized research participant recruiting agency. Participants with injuries, those with blood pressure higher than 140/90, those unable to walk or using walking aids, or those unable to read and understand and sign the informed consent were excluded from this study. Participants’ characteristics are summarized in Table 1.

This pilot study was conducted in wintertime in Eindhoven, The Netherlands. The entire study lasted three weeks: one baseline week and two PA and exercise intervention weeks. Participants were invited to visit our facilities three times: the first one on the intake day, the second one after a baseline week, and the third one after two weeks of intervention. Because this was an exploratory study we had no control group. However, a full week of baseline could serve as reference for the intervention.

### 2.2. Laboratory Examinations

At the beginning of the first visit, the volunteers were asked to sign the informed consent in case they still agreed to take part in our study. Afterwards, they filled in 7 questionnaires. At first, they filled in the American Heart Association/American College of Sports Medicine Pre-Participation Questionnaire (AAPQ) [33] as a screening for cardiovascular risk and the Physical Activity Readiness Questionnaire (PAR-Q) [34] to understand their readiness to start a PA and exercise program. Brachial blood pressure (BP) (Omron M10-IT) was also measured to check if participants fit the inclusion criteria. Once it was confirmed that they could safely exercise, they were asked to fill in the additional 5 questionnaires. The Short Form-12 Survey (SF-12) was used to inspect the participants’ general health state [35]. The SF-12 is composed of physical and mental factors. The Health Survey [35,36] and its 12 item short version is a reliable tool to assess health status [37]. Its reliability has been confirmed also in independent living older adults [38]. The degree of fatigue that the older adults may have experienced was examined by the Fatigue Severity Scale (FSS) [39,40]. The fatigue severity scale (FSS) was originally developed in rheumatoid arthritis patients [39,41], and it can reproducibly assess fatigue in older adult individuals as well [42]. Attitudes towards activity pacing can now be monitored thanks to a recently developed and validated questionnaire, as also used in Abonie et al. [21]. The Activity Pacing Questionnaire (APQ) examined their attitudes towards PA pacing strategies, with a specific focus on PA. It is composed of 7 items: 5 to evaluate the engagement in pacing and 2 to assess the perceived risk of over-activity [43]. Exercise self-efficacy (ESE) was measured using Bandura’s ESE scale [30,44]. Bandura [30] developed a tool to purposely quantify self-efficacy, and this has been deployed in older adults finding that PA predicted self-efficacy, although not as much as age and gender [45]. Finally, the Short Questionnaire to Assess Health Enhancing Physical Activity (SQUASH) was provided to the participants to evaluate their self-reported PA level [44,46]. In the common case in which PA monitoring technology would not be available, the SQUASH is often used to gather PA levels information from different weekly activities [44,46,47]. The SQUASH has acceptable internal validity and accuracy [47], even though it is known to suffer from overreporting [48]. All questionnaires were administered in Dutch, the native language of the participants. These were all validated translations, except for the AAPQ translated by a fellow researcher, a Dutch native speaker with a PhD degree as level of education.

During the first visit, participants received a wristwatch activity monitor (Philips Health Watch), and a tablet (Lenovo TAB4 10) on which the PACE mobile app was pre-installed. Participants were familiarized with the devices and explained how and when to re-charge them. The participants were asked to bring the devices home and use the activity monitor during daytime. During the baseline week the participants could not see any feedback on the PACE app and were asked to keep their normal physical activity behavior. During the second visit, participants’ body mass (Tanita InnerScan, Tokyo, Japan) and height (Seca W60092, Chino, CA, USA) were measured. The APQ and the SQUASH questionnaires were administered again. Heart rate (HR) and oxygen consumption (VO_2_) were recorded by a wearable metabolic system (Cosmed K5, Rome, Italy), while the participants were seated quietly for 5 min and during exercise. Participants’ exercise capacity was estimated by means of a six minute walk test (6-MWT). The 6-MWT is a validated submaximal CRF test, developed in chronic heart failure and chronic obstructive pulmonary disease [49], which was extensively used in clinical trials and validated in patients with different type of chronic diseases [50]; it was shown to be a valid and accessible protocol to assess CRF level in older adults [51]. This test has also been used in other age groups, including younger individuals [52]. The 6-MWT was performed along a 50 m long corridor. CRF was estimated according the following formula, developed by Kervio et al. [48]:CRF (VO_2max_ (mL·min^−1^)) = 2830.6 − (45.2 × age (yr.)) + (4.70 × BW (kg)) + (12.3 × height (cm)) + (1.75 × distance (m)) + (0.309 × VO_2_ (mL·min^−1^)) − (12.4 × HR (beat·min^−1^)).(1)

The reported accuracy of the CRF estimation based on Kervio’s equation is 97% [53].

After the test, participants received an explanation regarding their walking and exercise program and how to plan it using the PACE application. After the two weeks of intervention period, the participants visited the laboratory for the third and last visit. Body mass was measured again; then they were asked to fill in the APQ, EXE, FSS, SF-12, and SQUASH questionnaires, and they repeated the resting assessment and the 6-MWT. Finally, they provided unstructured feedback about the PACE mobile application usefulness and usability. This was arranged by the research team as unstructured interviews mainly aiming at understanding what the participants thought about the application, the wearable device, and the exercise program.

### 2.3. Physical Activity Intervention

The physical intervention was structured in two parts: a walking program, considered as moderate-intensity aerobic physical activity; and an exercise program, constructed using target HR, also referred to as exercise HR (HR_exercise_). HR_exercise_ was calculated with the heart rate reserve (HRR) method [32]. The HR_max_ was estimated using Tanaka’s formula for sedentary people (=211 − 0.8 × age) [54]. The weekly program was set according to the PA for Americans and WHO guidelines [1,2]. These consist of 30 min/day on five days per week of walking and 25 min/day on three days per week of exercise at 50–80% HRR. Although, the participants were instructed to perform their bouts of at least 10 min, only walking bouts shorter than 2 min ± 10 s were not counted as walking activities by the watch and were consistently not shown in the PACE app. One participant, with history of asthma, was prescribed with the walking program only. Participants’ progress was also viewed by means of a professional interface by the exercise physiologist in charge of the data collection. This allowed the prompt updating of the program for the second intervention week.

### 2.4. Adherence

In this study, we used two different definitions of adherence: program adherence and volume adherence. The users were exposed only to the program adherence. However, in the offline data analysis we were interested in comparing an alternative definition of adherence, volume adherence. At the same time, the PA program of the PACE app was divided in two elements: the walking program and the exercise program. Adherence to those two elements was also studied separately.

Walking program adherence was defined as the number of days in a week on which a participant reached the target walking duration (e.g., 30 min) divided by the target walking frequency (e.g., 5 days). Walking volume adherence was defined as the number of walking minutes per week divided by the target walking volume per week (e.g., 30 × 5 = 150 min). For exercise, similar definitions are used; the exercise program adherence focused on the number of days completed, whereas the exercise volume adherence focused on the number of minutes.

As mentioned above, the PACE app used the walking and exercise program adherence definitions. For each day it showed whether or not the daily walking and exercise targets had been reached. If not, the app showed the fraction of the daily walking and exercise target that had been reached. PACE did not take into account when a user walked more than the daily walking target, nor whether they had exercised more than the daily exercise target. In other words, PACE did not reward over-activity. Nonetheless, PACE acknowledged the unplanned days when participants reached their daily target.

### 2.5. PACE System

The PACE application is a research application designed to visualize patient’s adherence to PA programs. The PACE app ran on a tablet and received walking and exercise minutes from a wrist watch (Philips Health Watch) via Bluetooth Low Energy. The synchronization was automatically initiated when opening the app. The researcher (an exercise physiologist) entered the target walking minutes and the target heart rate zone and exercise minutes via a professional dashboard. PACE was designed for patients and older adult participants who often are less acquainted with technology. The aim of the system is to stimulate people to adhere to moderate-intensity activity (walking) and higher-intensity activity (e.g., brisk walking, biking, jogging) programs. PACE was delivered in Dutch [17].

On the main screen of the PACE app, which functioned also as a home screen, the participants could see their week progress, from Monday to Sunday. If they were engaged in both walking and exercising programs, they would see on the top half of the screen seven progress circles of the walking program and on the bottom half seven progress circles of the exercise program. The two halves were colored differently (e.g., blue and green), to stress the difference between the two programs, and were also differentiated by two different icons. All circles were empty at the start. However, the planned days stood out in bold. The circles would fill if the participants were either brisk-walking or exercising. When the participants would achieve the daily target, a white check mark appeared on the circle of that day. In case the daily target was be achieved in a non-training day, an opaque check mark appeared to stress the difference between planned days and non-planned days. If the participants exceeded the daily target, no further reward was provided.

### 2.6. Statistical Analysis

The statistical analysis was performed using SPSS (IBM Nederland B.V., Amsterdam, The Netherlands) and RStudio (Integrated Development for R. RStudio, PBC, Boston, MA, USA). The level of statistical significance was set at *p* < 0.05 (two-tailed) for all analyses. Data were presented as means ± standard deviations, unless otherwise noted. For each data collected, the Gaussian distribution was evaluatwith the D’Agostino–Pearson test. Two levels of repeated measures variables were analyzed with a paired *t*-test or the Wilcoxon signed-rank test if data sets were not normally distributed. Three levels of repeated measures variables were analyzed using one way-ANOVA for repeated measures or the Friedman test, if data sets did not assume the Gaussian distribution. In case there were violations of the sphericity, the corrections of Green-house Geisser were applied. Two mixed model ANOVA was used for the 6-MWT time × group analysis. Subgroups were determined in one case with a simple mean-split and in all other cases using the output of the cluster analysis. Recursive feature exclusion was executed for feature selection using the boruta R package [55], which is based on the random forest model. The selected variables were used as input for the κ-means cluster analysis. Number of clusters were empirically set to two, also in view of the significant result for the mean-split. Correlations were calculated using the Pearson’s coefficient.

## 3. Results

None of the demographic variables changed significantly during the intervention (Table 2). Additionally, APQ, FSS, SF-12, and PCS-12 scores did not change significantly. Self-reported minutes of activities collected with the SQUASH questionnaire did not change during the intervention period. Only the ESE score showed a significant main effect of time (Figure 1).

### 3.1. Program and Volume Adherence

No significant effect of time was found in exercise and walking adherence (for both adherences both program adherence and volume adherence). Adherence to both programs did not significantly increase or decrease during these two weeks when the participants were analyzed as one group. (Table 3; two weeks’ adherence to the walking and exercise programs).

As walking and exercise programs were kept separate, we evaluated how they related to one another by means of Pearson’s r correlations. Significant correlations were found between walking program adherence and exercise program adherence in the second intervention week (*n* = 7, *r* = 0.67, *p* = 0.049) and between walking volume adherence and exercise program adherence also during the second intervention week (*n* = 7, *r* = 0.77, *p* = 0.015).

Age had a significant level of correlation with walking program adherence (*n* = 8, *r* = 0.72, *p* = 0.019), and it showed a trend with exercise program adherence (*n* = 7, *r* = 0.60, *p* = 0.05). The 6-MWT showed a significant correlation with exercise program adherence (*n* = 7, *r* = 0.69, *p* = 0.040). A significant correlation was present between the physical component of the 12-item Short Form Health Survey (PC-12) and walking program adherence (*n* = 8, *r* = 0.634, *p* = 0.049).

### 3.2. Profiling

To select the independent variables (i.e., influencing factors) that best fitted the walking program adherence and the exercise program adherence, the recursive features elimination using the random forest model was executed. When walking program adherence was set as a dependent variable, the physical component of the SF-12 (PCS-12) was found to be the most predictive variable (Figure 2). However, when exercise program adherence was set as the dependent variable, the perceived risk of over-activity score from the APQ and HR_rest_ were found to be the most predictive variables (Figure 3). After that, a κ-means clustering algorithm was run using the PCS-12 to cluster for walking program adherence and the perceived risk of over-activity score and HR_rest_ to cluster exercise program adherence. The number of clusters was arbitrarily set to two, as the mean split of the data showed that this would be a sound choice. Walking program adherence clusters included four and six participants. Cluster one included four active participants, and cluster two included six less active participants (Figure 4). The cluster convergence was reached at the second iteration. This cluster analysis confirmed the validity of the selected variable. Exercise volume adherence clusters included five participants who exercised more (cluster one) and four participants who exercised less (cluster two) (Figure 5). In this case as well, cluster convergence was reached at the second iteration.

The 6-MWT distance did not show a statistically significant change over the two weeks of intervention at the group level (Figure 6). When this was mean split at baseline (faster *n* = 4, slower *n* = 6) it did result in a significant time × group interaction (F (1;8) = 14.05, *p* = 0.006). No main effect of time was found (F (1;8) = 0.00, *p* = 0.99), but there was main effect of group, and follow-up showed a significant simple main effect at baseline (t (8) = 5.57, *p* = 0.005) and at the end of the study (t (8) = 6.59, *p* < 0.001). A significant simple effect of time in the slower group (t (5) = 2.99, *p* = 0.03) was observed (Figure 7). When the distance covered during the 6-MWT was analyzed, using the two sub-groups identified by the walking program adherence κ-means cluster analysis (more and less active), it did not show any time × group interaction (F (1;8) = 3.04, *p* = 0.12). No main effect of time (F (1;8) = 0.34, *p* = 0.86), nor of group (F (1;8) = 0.48, *p* = 0.51) (Figure 8). Instead, when 6-MWT distance was analyzed using the two sub-groups derived from the exercise volume adherence κ-means clustering (more and less exercised), a significant time × group interaction (F (1;7) = 17.77 *p* = 0.004) was found. No main effect of time (F (1;7) = 1.20, *p* = 0.31), but a main effect of group (F (1;7) = 19.2, *p* = 0.003) was found. The follow-up showed groups difference at baseline (t (7) = 3.8, *p* = 0.007) as well as at the end of the study (t (5.2) = 5.26, *p* = 0.003). Furthermore, a significant decrease in 6-MWT distance covered over after two weeks was found in the less exercised group (t (3) = 3.99, *p* = 0.03) (Figure 9).

### 3.3. Qualitative PACE App Usability

The PACE application seemed to be well accepted by all 10 participants. This was evaluated by unstructured interviews. Thus, no objective evaluation could be made on its actual acceptance. There were, however, a few remarks. One participant reported over 10 minor bugs (e.g., some messages delivered in English instead of Dutch). One participant did not understand the synchronization feedback. During the 3 weeks of the study, two device malfunctioning events were noticed, because two participants accidentally unpaired their watches from the PACE app and thus activity data was no longer recorded by the app. This issue was solved by the researcher the following day. Two participants, who started the program in the middle of the week, did not like that they could not start the program from a set day; they had to wait for the upcoming Monday to start the intervention. Two participants spontaneously reported suggestions to improve the application (e.g., more motivational messages and images). Although all participants were shown how to plan walking and exercise days, this feature was actively used by 5/10 participants, who found it useful. Three participants underlined that the use of the app was an incentive to change their PA behavior (e.g., taking the stairs instead of the lift, walking to the local store instead of taking the car).

## 4. Discussion

### 4.1. Adherence

The physical component of the self-reported health state was shown to predict walking program adherence, while perceived risk of over-activity and HR_rest_ were predictors of HR-based moderate exercise program adherence.

In general, 50% of the apparently healthy older adult participants included in this pilot study did not meet the WHO PA guidelines, even during the intervention. The general health state, evaluated with the SF-12 questionnaire, showed a statistically significant correlation with walking program adherence only in the physical component. Both physical and mental component scores recorded in our pilot were higher when compared with those reported in the literature [35,36].

Activity pacing was shown to influence exercise program adherence if the participants perceived the risk of over-activity related to physical activity. This preliminary result seems to suggest that risk of over-activity could be a very useful profiling item when designing coping strategies for increasing adherence. It is also interesting to notice that this latent variable, perceived risk of over-activity, is relevant even at a low level of fatigue complaints. The level of fatigue in our small sample was low when compared to a similar population with a larger sample size [42]. Indeed, only one subject reported a fatigue score >4, which is the threshold used to detect the presence of fatigue [42].

It is indicative that the self-reported PA showed a poor correlation with measured PA. The participants overestimated their PA. These results confirmed the gap between self-reported and objectively measured PA [56]. In contrast to what was reported by McPhee et al. [13], self-efficacy was not strongly related with PA levels. Self-efficacy showed a significant decrease over time. Probably, the participants acquired more self-awareness and became more realistic about what they would be prepared to do to becoming more active. Indeed, individuals who already engaged in PA programs had higher self-confidence than whose had not yet begun to exercise regularly [57].

Although no control group was present, walking adherence seemed to increase thanks to the intervention (both program adherence and volume adherence). Yet, the volume adherence decreased by 4% from between the first week and second week of intervention, while program adherence increased by 6%. Thus, the program adherence definition seems to stimulate participants not to over-do activity during the week. Exercise adherence was more than twofold greater than walking adherence (referred to both program adherence and volume adherence). This could have several explanations. First of all, the exercise goal in minutes and days in a week was lower. The target HR was set based on the 6-MWT’s outcome, so that the participants could exercise by means of brisk walking. In addition, unlike walking minutes, exercise minutes were counted even if exercise bouts were shorter than two minutes. Hence, very short HR increases above threshold could had been recognized as exercise sessions. Four participants mentioned that the bad and cold weather (i.e., the study took place in wintertime) was a barrier to reach their PA and exercise targets. Winter season and adverse weather are known to have a negative effect on PA levels [56]. As was raised in Albergoni et al. [15], we stress here the importance of defining adherence consistently, and in addition to that, the definition of adherence should take into account users’ motivation and self-efficacy. Future research should focus on these aspects.

### 4.2. Profiling

From the cluster analysis, it emerged that the physical component PCS-12 of the SF-12 questionnaire was a good predictor of walking adherence. This confirmed that physical health state is a good PA level predictor [10]. Perceived risk of over-activity (APQ), and a low HR_rest_ were found to be good predictors of exercise adherence. These two variables did clearly cluster the participants into more exercised and less exercised, and a similar divide as shown for the mean-split was achieved by this data driven approach. The relation between lower HR_rest_ and a higher physical function is known in non-medicated healthy older adults [58]. However, this is the first time that perceived risk of over-activity has been associated with exercise adherence. This cluster analysis had the goal to identify PA and exercise adherence profiles based on baseline data. The final aim of these users‘profiles will be to personalize the program and its coping strategies to promote program adherence. It has been shown that personalized programs were more effective to improve the health state than non-personalized ones [59]. Although at a group level, the PA levels increased significantly during the first week of the program, 6 out of 10 participants had a reduced distance covered during 6-MWT. Those participants had already a low performance at baseline. We also observed that those six participants, although younger, had more medication use, as can be seen in Table 1. The split of our data into two groups was evident, and we decided to arbitrarily mean-split them into two groups. When we analyzed the changes of these two subgroups in time, we found a significant interaction. This meant that the participants who had a better 6-MWT outcome at baseline were the same ones who increased their performance after two weeks, and vice versa. The same time by group analysis was conducted by using two clustering models. One model was based on the independent variable explaining better walking adherence, namely PC-12. The second model was based on the two variables that explained somewhat better the adherence to the exercise program, namely the perceived risk of over-activity and HR_rest_. Only the second model was able to replicate the same significant interaction observed for the mean-split. We believe that the perceived risk of over-activity could be potentially used not only to predict adherence but also to tailor motivational and coping strategies to increase or maintain high adherence.

### 4.3. General Consideration

Our unstructured interviews at the end of the study revealed that the PACE application was overall well accepted by all participants. The monitoring and interfacing technologies may be a useful tool to promote PA adherence. However, personalization must be included in the final offering. With regard to long term adherence, it is difficult to predict whether personalization per-se could maintain high adherence. Gamification may be a possible strategy to stimulate it [60]. As already reported in the literature, there is a poor homogeneity to evaluate PA adherence [15]. In this study we included two different ways of calculating the walking and exercise adherence. We believe that adherence, as assessed by the PACE app, may better evaluate the fair PA intervention execution, because adherence defined as weekly total minutes against target weekly minutes does not consider activity load distribution, and it could induce over-activity at the end of the week.

### 4.4. Limitations

This study had several limitations. To begin with, this was a pilot study with an exploratory aim. This study did not have a control group. The small sample size and the short intervention period does not allow for strong definite conclusions. Moreover, because the intervention was not excessively demanding and the participants were not particularly unfit, we did not expect to observe a significant increase in their fitness at the group level after only two weeks of intervention. However, it does provide some interesting questions for future studies. For example, could profiling at baseline be used to motivate older adults to adhere to a walking and exercise program? Would people get bored after a few weeks? How can they be kept engaged? All these questions would require longer experimental studies. We are aware that the short nature of our study most likely allowed for higher adherence in using the app and in engaging in the PA program. Moreover, the three laboratory appointments the research team had with the participants during the three weeks of the entire study most likely increased participant engagement. In fact, we strongly advocate against using mobile technology to fully replace physical appointments with health professionals, but rather to complement, facilitate, and integrate those physical appointments. It was not the aim of this study to accept the hypothesis that purposed monitoring and user-interface technology improve adherence. This pilot study was aiming more at adding a small piece of knowledge of how such technology could be personalized to have a fair chance to be successful in its intent.

## 5. Conclusions

We provide here some preliminary evidence of what factors could influence PA program adherence in older adults. Subjective ratings of the physical component summary of the 12-item Short Form Health Survey strongly correlated with walking minutes as measured by a wrist-based activity monitor, and this same physical component was the best predictor of adherence to the walking program. Furthermore, HR_rest_ and perceived risk of over-activity were able to predict adherence to HR-based exercise program. These two factors could already at baseline cluster the group of healthy older adults into two groups able to predict improvements in the 6-MWT. Overall, the PACE app seemed to be well received by the older adults included in this study. Future studies should focus on larger samples, longer durations, and the use of user profiling to improve adherence.

## Figures and Tables

**Figure 1 ijerph-17-06142-f001:**
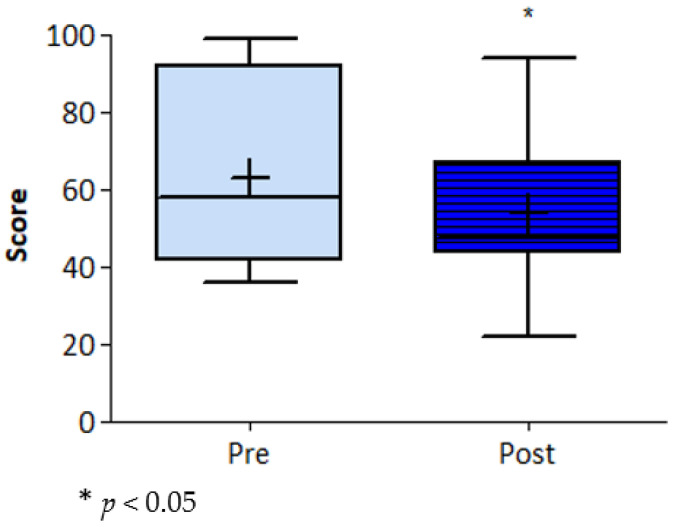
Exercise self-efficacy score.

**Figure 2 ijerph-17-06142-f002:**
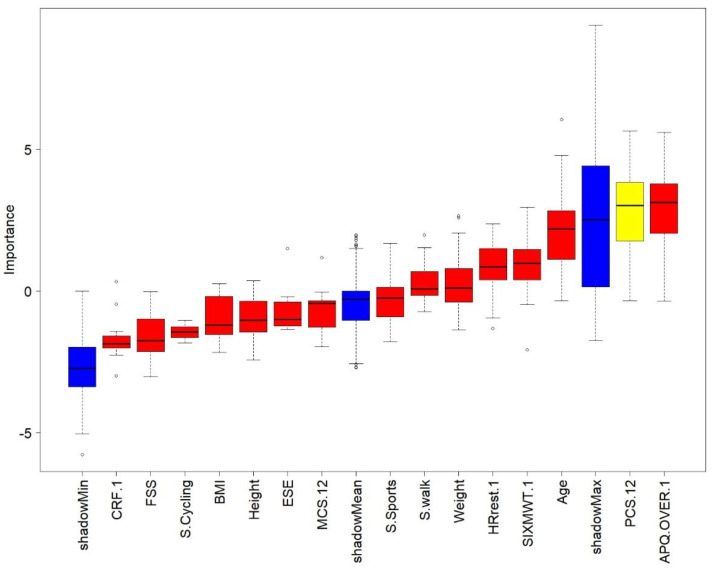
Recursive features elimination, walking minutes. Red color means that the variable is not considered important in predicting walking minutes. Yellow color means that the variable could be important in predicting walking minutes; however, there is a degree of uncertainty left. Blue color distinguishes the randomized copy variables used to assess importance.

**Figure 3 ijerph-17-06142-f003:**
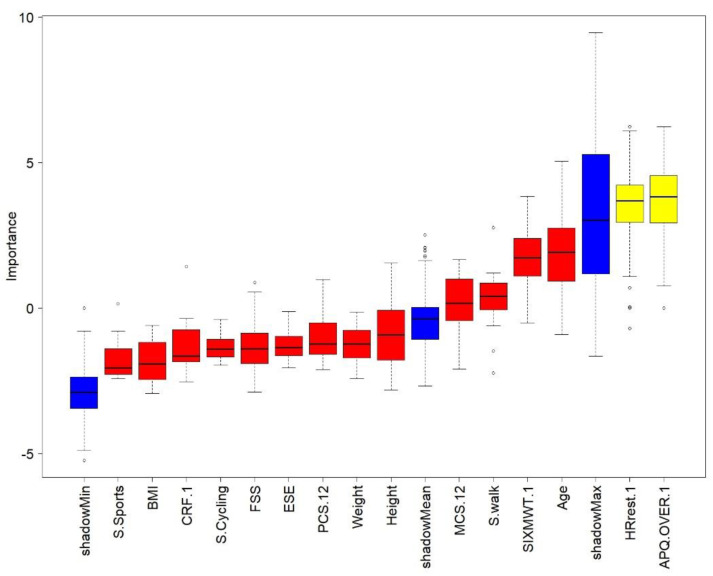
Recursive features elimination, exercise adherence. Red color means that the variable is not considered important in predicting exercise adherence. Yellow color means that the variable could be important in predicting exercise adherence; however, there is a degree of uncertainty left. Blue color distinguishes the randomized copy variables used to assess importance.

**Figure 4 ijerph-17-06142-f004:**
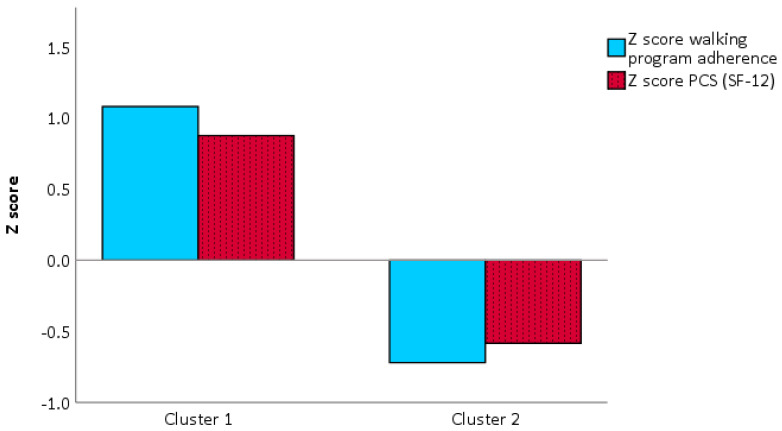
Walking minutes clusters.

**Figure 5 ijerph-17-06142-f005:**
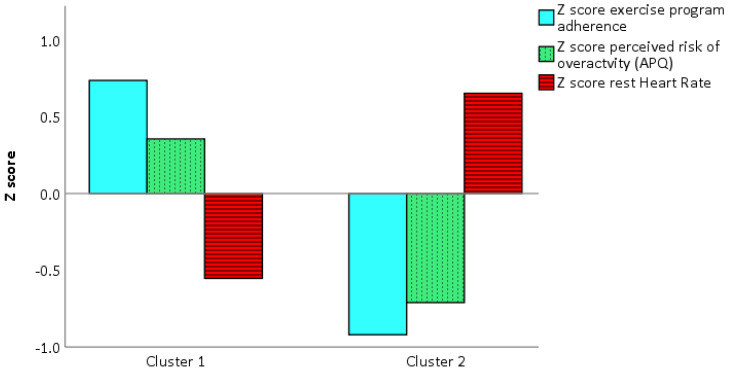
Exercise adherence clusters.

**Figure 6 ijerph-17-06142-f006:**
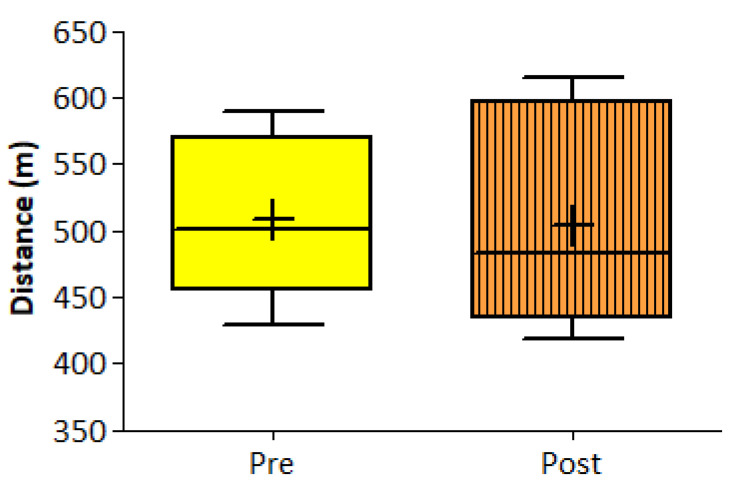
The 6 min walk test. Box plots of pre- and post-intervention distances covered during the 6MWT by the entire group.

**Figure 7 ijerph-17-06142-f007:**
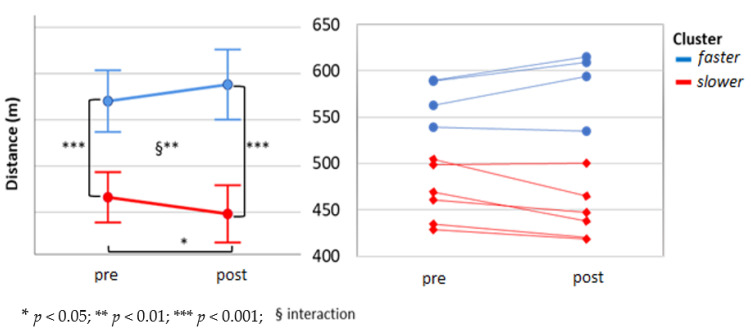
The 6-MWT mean split clusters. Two clusters were formed by using a simple mean split of the baseline distance covered during the 6MWT. Cluster 1, in blue, represents the faster sub-group, and cluster 2, in red, the slower group.

**Figure 8 ijerph-17-06142-f008:**
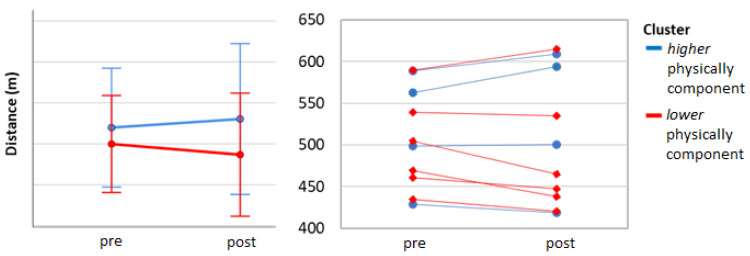
The 6-MWT walking program adherence clusters. Two clusters were yielded from the κ-means clustering method using the walking program adherence as input for recursive feature elimination (based on PCS-12). Cluster 1, higher physical component, in blue; and cluster 2, lower physical component, in red.

**Figure 9 ijerph-17-06142-f009:**
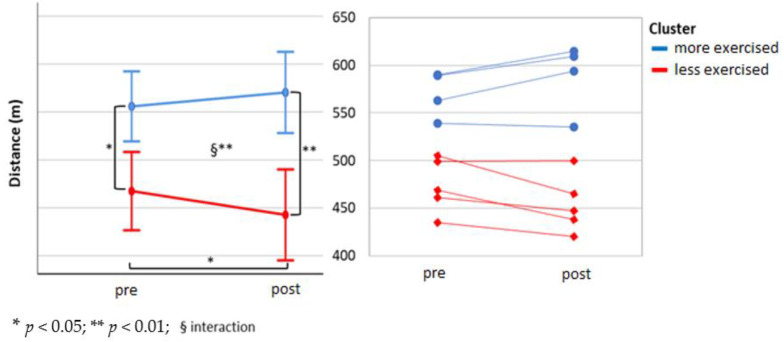
The 6-MWT exercise program adherence clusters. Two clusters were yielded from the κ-means clustering method using the exercise volume adherence as input for recursive feature elimination (based on APQ, HRrest). Cluster 1, more exercised, in blue; and cluster 2, less exercised, in red.

**Table 1 ijerph-17-06142-t001:** Participants’ characteristics at baseline.

Variable	Mean ± Standard Deviation		
	All	Faster (*N* = 4)	Slower (*N* = 6)
Age (yr.)	73.8 ± 2.3	74.3 ± 2.1	73.5 ± 2.6
Sex (male/female)	6/4	3/1	3/3
Height (m)	1.70 ± 0.11	1.68 ± 0.03	1.72 ± 0.14
Body mass (kg)	81.6 ± 13.8	78.4 ± 7.4	83.7 ± 17.3
BMI (kg/m^2^)	28.0 ± 2.0	27.8 ± 2.6	28.1 ± 1.8
Resting heart rate (bpm)	73 ± 13	65 ± 5	78 ± 15
Estimated absolute CRF (mL/min)	2086 ± 353	2144 ± 213	2047 ± 438
Estimated relative CRF (mL/kg/min)	25.7 ± 2.8	27.5 ± 2.9	24.5 ± 2.3
Blood pressure lowering medications (n/tot)	5/10	0/4	5/6
Cholesterol lowering medications (n/tot)	3/10	1/4	2/6
Smokers	2/10	1/4	1/6

BMI: body mass index, CRF: cardio respiratory fitness. Faster and slower were split according to the mean split of the distance covered during the six-minute walk test at baseline, as indicated in the Results section in the paragraph entitled Profiling.

**Table 2 ijerph-17-06142-t002:** Baseline, pre-, and post-intervention values.

Variables	Baseline Visit	Pre-Intervention Visit	Post-Intervention Visit	Pre vs. Post *T*-Test (df)	Mean Difference (95% CI)	*p*-Value	ANOVA F (df_t_; df_e_)	*p*-Value
Body Mass (kg)		81.6 ± 13.8	81.4 ± 13.8	0.83 (9)	0.20 (−0.34, 0.74)	0.46		
BMI (kg/m^2^)		28.0 ± 2.0	27.9 ± 2.1	0.81 (9)	0.06 (−0.11, −0.24)	0.44		
Resting heart rate (bpm)		73 ± 13	73 ± 11	^ᵂ^	^ᵂ^	0.88		
6-MWT ^a^ (m)		508 ± 60	504 ± 79	0.49 (9)	3.62 (−13.0, 20.2)	0.63		
Estimated CRF ^b^ (mL/kg/min)		25.7 ± 2.8	25.1 ± 3.0	1.28 (9)	0.61 (−0.47, 1.69)	0.23		
APQ ^c^								
Engagement in pacing	2.68 ± 0.92	2.86 ± 0.73	2.98 ± 0.75				^∯^	0.36
Perceived risk of over-activity	2.45 ± 0.98	3.00 ± 0.82	2.90 ± 0.84				2.28 (2; 18)	0.13
ESE ^d^	62.7 ± 23.6		53.7 ± 20.6	2.72 (9)	9.01, (1.52, 16.50)	0.024 *		
FSS ^f^	2.50 ± 0.77		2.76 ± 1.02	1.20 (9)	−0.26 (−0.74, 0.23)	0.26		
SF-12 ^g^ PCS-12 ^h^	53.5 ± 3.2		53.6 ± 4.5	^ᵂ^	^ᵂ^	0.95		
MCS-12 ^i^ (9/10 ^θ^)	58.4 ± 2.1		56.0 ± 3.4	2.26 (8)	2.38 (−0.05, 4.80)	0.05		
SQUASH ^j^								
Walking minutes	153 ± 180	126 ± 134	199 ± 324				^∯^	0.81
Cycling	136 ± 56	147 ± 163	101 ± 128				2.20 (2;18)	0.14
Other sports	120 ± 108	97 ± 65	90 ± 92				1.19 (2;18)	0.28

^a^ Six minutes walking test, ^b^ CARDIO respiratory fitness, ^c^ Activity Pacing Questionnaire, ^d^ exercise self-efficacy, ^f^ fatigue severity scale, ^g^ Short Form, ^h^ physical component summary scale, ^i^ mental component summary scale, ^j^ Short Questionnaire to Assess Health Enhancing Physical Activity, * *p* < 0.05, ^ᵂ^ Wilcoxon signed rank test performed, ^∯^ Friedman test performed, ^θ^ outlier removed.

**Table 3 ijerph-17-06142-t003:** Two weeks’ adherence to the walking and exercise programs.

Adherence	Baseline Week	1st Intervention Week	2nd Intervention Week	ANOVA F (df_t_; df_e_)	*p*-Value
Walking program adherence (%)	32 ± 39	48 ± 38	54 ± 38		0.38
No. participants reaching the walking target	1/10	2/10	2/10		
Walking volume adherence (%)	66 ± 58	108 ± 63	104 ± 60	3.5 (2; 18)	0.05 *
No. participants reaching the target walking volume	3/10	4/10	4/10		
Exercise program adherence (%)	96 ± 82	115 ± 99	111 ± 78	0.44 (2;16)	0.65
No. participants reaching the exercise target	5/9	5/9	6/9		
Exercise volume adherence (%)	263 ± 232	252 ± 181	260 ± 216	0.29 (2; 16)	0.97
No. participants reaching the target exercise volume	7/9	6/9	6/9		

* *p* < 0.05.

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
