# Peer review of "Factors Influencing Walking and Exercise Adherence in Healthy Older Adults Using Monitoring and Interfacing Technology: Preliminary Evidence"

_ijerph, 2020, doi:10.3390/ijerph17176142_

Round 1

Reviewer 1 Report

The authors present an interesting account of a mobile app-delivered walking and exercise adherence programme. The research is interesting, and the manuscript would benefit from some improvement, as explained below.

A general point regarding terminology: I would advise against using the term ‘elderly’, as it is seen as a loaded term, conjuring up images of frailty and dependence. See this article for a discussion of the issue: https://digitalcommons.sacredheart.edu/cgi/viewcontent.cgi?article=1157&context=pthms_fac

I would use ‘older adults’ instead; it’s much safer. Please check your manuscript and replace the term throughout.

Introduction:

The authors have done well to set the PA and technology context for the research in the introduction. The intervention is described well, along with the factors to be considered.

Line 62-3: I would write that tech-based exercise interventions ‘seem to promote’, or ‘seem to demonstrate good adherence’, rather than ‘show a good adherence’.

Line 86: replace ‘the way how elderly people perceive’ with the way that older adults perceive’.

Line 100: ‘enrolling’ not ‘enrroning’, I think?

Line 107: Do the authors mean to use the term ‘inflicting’ here? Or do they mean ‘influencing’?

Methods:

The authors have described the elements of the intervention, and the various measures employed, in clear detail.

P118: Can the authors please explain what a recruiting agency is? Was that part of the university, a health service? A leisure service? In the UK, this term is used for employment agencies, so it needs some clarification please.

Line 119: ‘of’ not needed here – ‘using of walking aids’.

Table 1, row 5: ‘Hear’ should be ‘Heart’.

Line 155: ‘suffer from’, not ‘suffer of’.

Line 170: Can the authors please rephrase the sentence using reference 51? ‘health people’, should be ‘healthy people’ I think, but this implies that older adults are not healthy, which I would dispute. I suggest it is rephrased as something like ‘is used in other age groups, including younger individuals’.

Results and Discussion:

I have a concern about testing for significant changes with such a short time period and small numbers of participants. Surely one would not expect to see significant changes? The authors are presenting preliminary research here, so effect sizes have not been calculated (perfectly acceptable), so it is important to make this clearer, I think. The sentence on lines 414-415 should be expanded to explain the point.

There is also the issue of app/technology engagement to consider. When users download an app, they tend to engage with it well for a short time, before engagement tails off. The short term nature of this intervention may not give indicative results for longer term engagement. The authors should also discuss this issue in relation to their results. There is some literature on this (e.g. Krebs P Duncan DT Health App Use Among US Mobile Phone Owners: A National Survey JMIR Mhealth Uhealth 2015;3(4):e101 DOI: 10.2196/mhealth.4924) Finally, the intervention had a considerable amount of contact time, which could also affect engagement (highly likely I think). The authors should recognise and discuss this in the limitations.

Author Response

Reviewer 1

The authors present an interesting account of a mobile app-delivered walking and exercise adherence programme. The research is interesting, and the manuscript would benefit from some improvement, as explained below.

A general point regarding terminology: I would advise against using the term ‘elderly’, as it is seen as a loaded term, conjuring up images of frailty and dependence. See this article for a discussion of the issue: https://digitalcommons.sacredheart.edu/cgi/viewcontent.cgi?article=1157&context=pthms_fac

I would use ‘older adults’ instead; it’s much safer. Please check your manuscript and replace the term throughout.

Authors: We thank the reviewer to point out this issue to us. We were not aware of it. We have now replaced it throughout the manuscript as suggested by the reviewer.

Introduction:

The authors have done well to set the PA and technology context for the research in the introduction. The intervention is described well, along with the factors to be considered.

Line 62-3: I would write that tech-based exercise interventions ‘seem to promote’, or ‘seem to demonstrate good adherence’, rather than ‘show a good adherence’.

Authors: This is indeed a good suggestion, as it applies better to the fact that our statement refers to a systematic review, and not directly to experimental evidence. We have made the change.   

Line 86: replace ‘the way how elderly people perceive’ with the way that older adults perceive’.

Authors: We have made the change.

Line 100: ‘enrolling’ not ‘enrroning’, I think?

Authors: Thanks for spotting that. We apologize. We have rectified the typo.

Line 107: Do the authors mean to use the term ‘inflicting’ here? Or do they mean ‘influencing’?

Authors: Again thanks for spotting that. We should have proof read it a bit more carefully. We have rectified the typo.

Methods:

The authors have described the elements of the intervention, and the various measures employed, in clear detail.

P118: Can the authors please explain what a recruiting agency is? Was that part of the university, a health service? A leisure service? In the UK, this term is used for employment agencies, so it needs some clarification please.

Authors: Thank you for the questions. The recruiting agency is not an employment agency. It is a private company, which has a database of volunteers that are happy to take part in research projects.  The volunteers subscribe to those kind of agencies so that they can be informed when a new study is available. Volunteers are often pensioners, thus, ideal for studies focusing on the older population. The recruiting agency also compensates the volunteers for their participation. The service of a recruiting agency is to send the poster to the people in their database, which fit some basic criteria such as age, health status, etc. In case the volunteers were interested in our study, based on the poster, they were put in contact with our team via a functional account, and provided with the information letter by us. All this was strictly handled according to the GDPR. The recruiting agency charges a commission per recruitment enquiry, plus the compensation money for the participants. The advantage of making use of this service is that the participants are usually familiar with what taking part in a study means. Volunteers would be compensated even if they withdraw during the study. In order not to mix it up with the British employment agencies, we can call it: specialized research participant recruiting agency.   

Line 119: ‘of’ not needed here – ‘using of walking aids’.

Authors: Amended.

Table 1, row 5: ‘Hear’ should be ‘Heart’.

Authors: It should be indeed. Sorry! Corrected.

Line 155: ‘suffer from’, not ‘suffer of’.

Authors: Amended.

Line 170: Can the authors please rephrase the sentence using reference 51? ‘health people’, should be ‘healthy people’ I think, but this implies that older adults are not healthy, which I would dispute. I suggest it is rephrased as something like ‘is used in other age groups, including younger individuals’.

Authors: Indeed, that sentence this not read well. We have now changed it as suggested.

Results and Discussion:

I have a concern about testing for significant changes with such a short time period and small numbers of participants. Surely one would not expect to see significant changes? The authors are presenting preliminary research here, so effect sizes have not been calculated (perfectly acceptable), so it is important to make this clearer, I think. The sentence on lines 414-415 should be expanded to explain the point.

Authors: We appreciate that the reviewer has understood the philosophy of this preliminary work. We indeed do not want to oversell the outcomes of this study. In theory, significant changes in activity, and fitness related parameters, such as 6 min walked distance can be seen in two weeks. However, these effects would require i. a very heavy intervention and/or ii. a very low baseline. We did not recruit for particularly sedentary people and our intervention was not harsher than the guidelines. Thus, we agree with the reviewer that at a group level it would be no surprise not to see significant differences. We have tried to clarify this in the limitations.  

There is also the issue of app/technology engagement to consider. When users download an app, they tend to engage with it well for a short time, before engagement tails off. The short term nature of this intervention may not give indicative results for longer term engagement. The authors should also discuss this issue in relation to their results. There is some literature on this (e.g. Krebs P Duncan DT Health App Use Among US Mobile Phone Owners: A National Survey JMIR Mhealth Uhealth 2015;3(4):e101 DOI: 10.2196/mhealth.4924) Finally, the intervention had a considerable amount of contact time, which could also affect engagement (highly likely I think). The authors should recognise and discuss this in the limitations.

Authors: We agree with the reviewer, and in fact, we already raised these issues as questions. Now we have elaborated on these issues further in the discussion, under the limitations subsection.

Reviewer 2 Report

This work presents a study aimed at studying different potential predictors related to a Walking and Exercise adherence program in healthy elderly individuals.

The paper is generally well-written and easy to follow, although it may benefit from additional copy-editing.

I liked this paper, but I also have several comments that may help strengthen this work. The main identified weaknesses are the presentation, some clarity issues, and the overall contribution. I next describe in detail per section.

METHODS

In Table 1, can the authors be more specific about the last three (medication, blood pressure, cholesterol)? Were participants asked about medication and cholesterol? Perhaps more important, we may need two sub-groups in this table regarding (faster vs slower) participants’ performance. This can help understand better the results presented at the end of the paper.

In the Discussion section the authors mention that it was conducted during winter. However, we need this description in the Methods section.

It is unclear if the instruments were translated to Dutch by the authors, or they used a validated version. Please specify.

Regarding the Activity bouts (at least 10 min) in the PA Intervention subsection, Are consecutive, non-stop 10 minutes? For instance, 5 min, some rest, and then another 5 min did not count as a bout? This is just for the sake of clarity.

Since the mobile app is considered instrumental in this study, we may need a screenshot of the PACE app.

RESULTS

In Table 3, it is unclear what the mean values represent. For instance, according to the definition on the Adherence subsection in Methods, the Walking program adherence was the number of days reaching a specific number of minutes (say 30 min), divided by the number of target days. The arithmetic mean has to be less than a 1, since it is a ratio (or less than a 100 if taken as a percentage). The same goes for volume. As far as I understood, since the PACE app did not reward over-activity, in case there was over-activity, this was not considered for analysis. Or was it? If it was considered for analysis, this is unclear. This is related to my next comment.

In the “Exercise volume adherence (%)” has values above the 100% (e.g., 263 ± 232 during the baseline week). Are these really percentages? Or are these minutes?

Also, the first line of the section Program and volume adherence seems odd to me by referring to the effect of time (“No significant effect of time was found in exercise and walking adherence” on page 7). Are you referring to the percentage of days (or minutes) for the walking and exercise programs? I know this is time, but the statement is vague.

On page 10, the subgroups obtained through the mean split, we need the means (descriptive data). The authors are showing the results of the simple main effects (pre vs post), but we need to see the mean values (and std. dev.) for the sub-groups. It’s shown in Figure 7, but in the text would also be useful.

On page 10, I did not understand this “No main effect of time (F(1;7)=1.20, P=0.31), but a main effect of group (F(1;7)=19.2, P=0.003).” Why the effect of time shows an F-ratio greater than 1? Isn’t this test supposed to last for 6 minutes per participant? Or were there some participants who did not complete the 6 minute walking test?

QUALITATIVE PACE APP USABILITY

The authors mention that “The PACE application was well accepted by all 10 participants.” How was this assessed? And analyzed? There is no report in the Methods section of interviews conducted with participants regarding “technology acceptance”. There are widely accepted quantitative instruments such as the TAM (Technology Acceptance Model), which is used to obtain participants’ perceptions on usefulness and ease of use of certain technology.

Also, there are quantitative ways to measure usability (efficacy, efficiency, usefulness, user experience). My main concern here refers to the claims made. Please include more details of the instruments used to collect data, the data analysis, and exact constructs that were measured. This is important, since it has to harmonize with what is claimed in the introductory sections: “Our rationale is that the use of monitoring and user-interface technology can promote adherence…” Then, is user interface technology important for adherence? From the evidence you are showing, I don’t think that conclusions can be made regarding the user interface, usability, or technology acceptance.

DISCUSSION

In the Profiling sub-section, “6 out of 10 participants had a reduced distance covered during 6-MWT.” Can you provide a plausible explanation for this? These differences are evident, and shown in the figures, but this paper would benefit from a more complete explanation of this.

In the General Consideration sub-section, there are some details of the type of interview you used (i.e., unstructured interview). We need this info in the Methods section. Also, read my comments above regarding this topic and the claims made about the relationship between adherence and user interface (or usability).

OVERALL CONTRIBUTION

I think this paper needs to better frame their contribution. The paper title, and some sections such as the Role of Technology subsection, and a few statements in the introductory section give some emphasis on the role of the mobile app in this work. However, the evidence collected in this regard is weak at best. Not much is (formally) collected pertaining to how the app influences adherence, which it seems it was one of the motivations of this work (“We have developed a mobile application with the specific aim of increasing PA program adherence.” on page 2). The paper title also places particular emphasis on the way the program was delivered (i.e., the mobile app), but again, little was carried out in this regard, apart from having he entire 10 participants receiving the same overall treatment (i.e., mobile app and intervention program).

I guess my point here is that the authors need to revise what the overall contribution is (which I believe they have), and harmonize the title, abstract, introductory paragraphs, methods, and conclusions (based on the evidence collected). For instance, the title refers to mobile app, but the abstract refers to wrist-based PA adherence, and describes the mobile app as an app to watch the progress (“Participants saw their progress via a purposely designed mobile application.”), however this was not looked into, or at least it is not shown. For this to happen, at the very least I would expect an analysis of how the elements of the user interface or the mobile app (i.e., the type of feedback provided, colors, timeliness) were a predictor for the walking/exercise adherence programs, which again, was apparently not looked into. Alternatively, the effect of the mobile app could have been observed though a control group, but this is not the case. Then, my suggestion is to revise title, abstract, a few sections of the introduction, and the overall claims made.

MINOR CONCERNS

  • Figure 2 and Figure 3 are blurry.
  • In Figure 2 and Figure 3, please include a brief explanation about the meaning of the colors red, blue, and yellow.

Author Response

Reviewer 2

This work presents a study aimed at studying different potential predictors related to a Walking and Exercise adherence program in healthy elderly individuals.

 The paper is generally well-written and easy to follow, although it may benefit from additional copy-editing.

I liked this paper, but I also have several comments that may help strengthen this work. The main identified weaknesses are the presentation, some clarity issues, and the overall contribution. I next describe in detail per section.

METHODS

In Table 1, can the authors be more specific about the last three (medication, blood pressure, cholesterol)? Were participants asked about medication and cholesterol? Perhaps more important, we may need two sub-groups in this table regarding (faster vs slower) participants’ performance. This can help understand better the results presented at the end of the paper.

Authors: Thanks for the comment. We understand that the information about the medication as reported in Table 1 is misleading. We have now amended the table so that the information is no longer misleading. Additionally we realized we omitted the information about the smokers, now added. We have added the information for the two subgroups faster and slower. This was indeed a good suggestion.    

In the Discussion section the authors mention that it was conducted during winter. However, we need this description in the Methods section.

Authors: This is a good point. In the very first version of the manuscript, we stated the months and the year when the data collection was executed in the methods section. However, I (FS) realized that stating the precise months, and year, would possibly increase the risk of breaching the participant’s data protection. Meaning that it could become easier to identify from the exact months and year of participation, the location of the study, age range and so on, the identity of the volunteers. Thus, I have removed this detail. Yet, I agree with the reviewer that it information, as “the study was conducted in wintertime” needs to be provided already in the methods session. Now amended.     

It is unclear if the instruments were translated to Dutch by the authors, or they used a validated version. Please specify.

Authors: Thanks for the opportune observation. We used validated translations, except for the AAPQ translated by a fellow researcher, Dutch native speaker, with a PhD degree as level of education. Now amended in the text.  

Regarding the Activity bouts (at least 10 min) in the PA Intervention subsection, Are consecutive, non-stop 10 minutes? For instance, 5 min, some rest, and then another 5 min did not count as a bout? This is just for the sake of clarity.

Authors: Our technology did have a hard filter set at 2 minutes. Meaning that any walking activity, which was not continuously performed for 2 minutes or longer, was not counted by the app. We did instruct the participants to engage in activity bouts of at least 10 minutes in conformity with the guidelines. However, the app would still count continuous activities even if shorter than 10 minutes as long as they were longer than the 2 minute threshold. This to avoid discouraging the participants. We tried to clarify this further in the text.  

Since the mobile app is considered instrumental in this study, we may need a screenshot of the PACE app.

Authors: We are sharing the screenshot of the app here below on confidential basis. Please do not share it further. The design of the app is protected by USD820881S1 patent. However, since the app is not commercially available, we cannot share the screenshot publicly.   

For the Screenshot Please see FILE attached

RESULTS

In Table 3, it is unclear what the mean values represent. For instance, according to the definition on the Adherence subsection in Methods, the Walking program adherence was the number of days reaching a specific number of minutes (say 30 min), divided by the number of target days. The arithmetic mean has to be less than a 1, since it is a ratio (or less than a 100 if taken as a percentage). The same goes for volume. As far as I understood, since the PACE app did not reward over-activity, in case there was over-activity, this was not considered for analysis. Or was it? If it was considered for analysis, this is unclear. This is related to my next comment.

In the “Exercise volume adherence (%)” has values above the 100% (e.g., 263 ± 232 during the baseline week). Are these really percentages? Or are these minutes?

Authors: We recognize to have caused this misunderstanding. We need to draw the reviewer’s attention on the screenshot of the app shared above. The PACE app did not reward for extraminutes performed in a given day. Yet, it did give credit to the user who reached the daily target in unplanned days (see the opaque check marks).Thus, for our calculation, we defined program adherence: “the number of days in a week on which a participant reached the target walking duration (e.g. 30 minutes), divided by the target walking frequency (e.g. 5 days)”, as described in the text. It is true, that in the introduction we state: “The users were able to decide what day to perform the prescribed training. Training longer, or more days than prescribed, was not rewarded by the app”. We realized that this statement is in fact not technically accurate. The part “or more days” is not in line with our calculation of adherence. Now we have removed that.  We have further clarified this point by adding; “Nonetheless, PACE gave credit to the participants who were also active on non-planned days.” We would like to thank the reviewer to make us double-check this aspect and so allowing us to spot this inconsistency.  

We double-checked the volume adherence calculations and we confirm that these were above 100% and were expressed them in percentage and not in minutes. For example for a subject we have: Weekly Target: 3x25’=75’ (100%). Baseline Week: 169’ (225%). Week 1: 210’ (280%). Week 2: 162’ (216%). In particular, with respect to exercise, several older adults in the Netherlands use the bike to go to the shops and other places, and these provided them with a higher exercise activity count.   

Also, the first line of the section Program and volume adherence seems odd to me by referring to the effect of time (“No significant effect of time was found in exercise and walking adherence” on page 7). Are you referring to the percentage of days (or minutes) for the walking and exercise programs? I know this is time, but the statement is vague.

Authors: Thanks to point out this, as is our intention to be clear. With this statement, we mean that the adherence to both programs did not significantly increase or decrease during these two weeks when the participants were analyzed as one group. We have added this statement.     

On page 10, the subgroups obtained through the mean split, we need the means (descriptive data). The authors are showing the results of the simple main effects (pre vs post), but we need to see the mean values (and std. dev.) for the sub-groups. It’s shown in Figure 7, but in the text would also be useful.

Authors: This is a good suggestion and fits with the reviewer’s first comment, as these are also the faster and slower, we have added this information in table 1.  

On page 10, I did not understand this “No main effect of time (F(1;7)=1.20, P=0.31), but a main effect of group (F(1;7)=19.2, P=0.003).” Why the effect of time shows an F-ratio greater than 1? Isn’t this test supposed to last for 6 minutes per participant? Or were there some participants who did not complete the 6 minute walking test?

Authors: The critical F for the given degrees of freedom (1,7) is 5.59, thus a 1.2 F justifies a non-significant effect of time (pre, post). In other words, F ratio does not correspond to the ratio of the means times of the 6 min test, but to the ratio between Mean Squared within conditions (time in our case)/ Mean Squared Error. Both Mean Squared within conditions and Mean Squared error are inversely proportional to their correspondent degrees of freedom. All participants completed the 6 min tests.

QUALITATIVE PACE APP USABILITY

The authors mention that “The PACE application was well accepted by all 10 participants.” How was this assessed? And analyzed? There is no report in the Methods section of interviews conducted with participants regarding “technology acceptance”. There are widely accepted quantitative instruments such as the TAM (Technology Acceptance Model), which is used to obtain participants’ perceptions on usefulness and ease of use of certain technology.

Authors: We agree with the reviewer. There are tools to better evaluate the acceptance of a technology. In our previous work we have used Computer System Usability Questionnaire (CSUQ) doi: 10.1080/10447319509526110, see Rospo et al doi: 10.2196/mhealth.5518. In this study, however, we had already a large number of questionnaires so we decided to run unstructured interviews, when closing the study. Another reason for doing so, was that the same app was extensively tested with the same target group, in terms of age in other studies (unpublished data), during the design and the development of the app itself.

The same app has been used in two other separate studies (Ensom E, Albuquerque D, Erskine N, Peterson A, Dickson E, Ding EY, Piche J, Wang Z, Escobar J, Alonso AA, Makam R. Feasibility of, and Adherence to, a Novel, Home-based Cardiac Tele-rehabilitation Program for Heart Attack Survivors: The MI-PACE Study. Circulation. 2019 Nov 19;140(Suppl_1):A11873-) and the second as part of the ATMoSPARE project ( https://www.imw.fraunhofer.de/content/dam/moez/de/documents/Jahresbericht_2015-16/Unternehmensentwicklung/E-Health-Innovationen%20f%C3%BCr%20mehr%20Lebensqualit%C3%A4t%20bei%20Multimorbidit%C3%A4t.pdf). The result of the second one are not yet published. We therefore had insights on the usability and acceptance of the app prior to the start of this investigation. Yet, we decided to have an unstructured interview to understand if the acceptance of the participants in this study was in line with the other studies.  

Also, there are quantitative ways to measure usability (efficacy, efficiency, usefulness, user experience). My main concern here refers to the claims made. Please include more details of the instruments used to collect data, the data analysis, and exact constructs that were measured. This is important, since it has to harmonize with what is claimed in the introductory sections: “Our rationale is that the use of monitoring and user-interface technology can promote adherence…” Then, is user interface technology important for adherence? From the evidence you are showing, I don’t think that conclusions can be made regarding the user interface, usability, or technology acceptance.

Authors: We again agree with the reviewer. The evidence we provide here cannot support that monitoring and user-interface technology is important for increasing adherence. It was not our intention to support claims with no evidence. The aim of this study was more too explore how influencing factors could be used to improve future personalization of such monitoring and interfacing technology. The question: does technology help adherence at all? is still open. In our systematic review and other recent ones it seems that this hypothesis makes sense. With our preliminary evidence we aim to add a small brick of knowledge toward building a supporting system making use of technology that may have a fair chance to test that very hypothesis.

DISCUSSION

In the Profiling sub-section, “6 out of 10 participants had a reduced distance covered during 6-MWT.” Can you provide a plausible explanation for this? These differences are evident, and shown in the figures, but this paper would benefit from a more complete explanation of this.

Authors: Our idea of why the 6 people who had a poor 6-min walk test at baseline did not improve, is that they were also less healthy, as matter of fact those are also the one on more medications. As now shown in Table 1 revised. We complimented this theory in the discussion.   

In the General Consideration sub-section, there are some details of the type of interview you used (i.e., unstructured interview). We need this info in the Methods section. Also, read my comments above regarding this topic and the claims made about the relationship between adherence and user interface (or usability).

 Authors: We do state in the methods: “Finally, they provided unstructured feedback about the PACE mobile application usefulness and usability”, we now tried to increase clarity.

OVERALL CONTRIBUTION

I think this paper needs to better frame their contribution. The paper title, and some sections such as the Role of Technology subsection, and a few statements in the introductory section give some emphasis on the role of the mobile app in this work. However, the evidence collected in this regard is weak at best. Not much is (formally) collected pertaining to how the app influences adherence, which it seems it was one of the motivations of this work (“We have developed a mobile application with the specific aim of increasing PA program adherence.” on page 2). The paper title also places particular emphasis on the way the program was delivered (i.e., the mobile app), but again, little was carried out in this regard, apart from having he entire 10 participants receiving the same overall treatment (i.e., mobile app and intervention program).

I guess my point here is that the authors need to revise what the overall contribution is (which I believe they have), and harmonize the title, abstract, introductory paragraphs, methods, and conclusions (based on the evidence collected). For instance, the title refers to mobile app, but the abstract refers to wrist-based PA adherence, and describes the mobile app as an app to watch the progress (“Participants saw their progress via a purposely designed mobile application.”), however this was not looked into, or at least it is not shown. For this to happen, at the very least I would expect an analysis of how the elements of the user interface or the mobile app (i.e., the type of feedback provided, colors, timeliness) were a predictor for the walking/exercise adherence programs, which again, was apparently not looked into. Alternatively, the effect of the mobile app could have been observed though a control group, but this is not the case. Then, my suggestion is to revise title, abstract, a few sections of the introduction, and the overall claims made.

Authors: We understand the point of view of the reviewer, as we also understand that the reviewer does not think that we are trying to oversell our results. We did design the PACE app with the aim of improving adherence, yet we do not test this claim here. This is certainly true. We agree that we need to make this more explicit. In line with the suggestion of the reviewer, we tried to de-emphasize the central position of the app in the title. Now: “Factors influencing Walking and Exercise adherence in healthy older adults using monitoring and interfacing technology: preliminary evidence”. Additionally we toned down the opening statement of the Abstract. Because of word count reasons it does not allow for extensive changes, in our view. We also added a clear statement that: “… this pilot study was not designed to accept or reject the hypothesis that the PACE app would be able to increase adherence, as this would require an adequately sized randomized control trial.”      

MINOR CONCERNS

  • Figure 2 and Figure 3 are blurry.
  • In Figure 2 and Figure 3, please include a brief explanation about the meaning of the colors red, blue, and yellow.

Authors: Added the explanation of the colors. In addition, we would like to mention that Boruta is an improvement of the Random Forest feature selection method, also known as recursive feature elimination, which to be fair can also be done with other methods such as logistic regression. Boruta algorithm adds randomness to the importance evaluation algorithm, let’s say the recursive feature elimination just mentioned, so that the certainty about the importance of a given variable is increased. In short, a randomized copy of the variables is made at each iteration of the random forest importance computation. Thus if a variable has a higher importance than the maximal importance of all randomized attributes, is retained (green), if there is some uncertainty (yellow), if it has a lower importance it is rejected or discarded (red). Blue represent the importance of the randomized copies in their complete distribution spectrum, that is: Min. Mean and Max. Resolution has been increased.  

Round 2

Reviewer 2 Report

I thank the authors for the amendments made to the manuscript. I have a few minor comments:

METHODS

In the Laboratory examinations section, please include the references of the validated instruments in Dutch language as well as the reference of the original instrument in English language.

If the screenshot of the app cannot be included, can you include a brief desription of the elements of the ser interface that are included in the app? This would provide the reader with information regarding the instrument used.

RESULTS

My suggestion regarding the results shown in Table 3 is sticking to the “adherence” definition. I mean, you want to know if they complied with the suggested amount of exercise (not more than that i.e., over 100%). I think you are not interested in finding whether there are differences in the amount of walking/exercise, but rather if they could comply with the suggested work plan (up to 100%). Or are you also interested if they could follow the precise amount of exercise? That is, more than the recommended is bad, and so is less than the recommended amount?

If you do not consider the “extra” walking/exercise, in the interventions (capping to 100%), I think some variables may yield significant differences, or it may be clearer to the reader. However, if you want to consider the “extra” walking/exercise time, then, the way is carried out at the moment is fine. This is not compulsory. It is just a suggestion for this paper or the next one. It depends on what you want to show. I guess this as the reason I was confused in the previous version.

MINOR CONCERNS

  • In Table 1, please be specific about the note at the bottom: “Faster and Slower are split according to the mean split”, particularly the mean split of what variable. Also, I would explain that the variable used for splitting the group relates to results found later in the manuscript.
  • “Expect” should be “except” I guess (page 4, line 161)
  • On page 5, the new text reads: “PACE gave credit to the participants who were also active on non-planned days.” What do you mean by giving credit? Recognition? In what way? You menioned in your answer that it was though the opaque check marks. Please include this explanation. As is, it is unclear.

Author Response

We would like to thank the reviewer for the attention to details that allowed us to improve further our work.    

METHODS

In the Laboratory examinations section, please include the references of the validated instruments in Dutch language as well as the reference of the original instrument in English language.

Authors: As a matter of fact, the only reference missing of the questionnaires used in Dutch is the one for the FSS. We added it.  Ref #40.

If the screenshot of the app cannot be included, can you include a brief desription of the elements of the ser interface that are included in the app? This would provide the reader with information regarding the instrument used.

 Authors: yes, we recognize that it is a disadvantage not to show the screenshot. However, it is indeed confidential. We are happy that the reviewer appreciates that. We added a description of the main screen of the app, which we shared in the previous round. We appreciate the reviewer’s advice because now the reader will have the chance to have a better understanding of how the app actually worked.

RESULTS

My suggestion regarding the results shown in Table 3 is sticking to the “adherence” definition. I mean, you want to know if they complied with the suggested amount of exercise (not more than that i.e., over 100%). I think you are not interested in finding whether there are differences in the amount of walking/exercise, but rather if they could comply with the suggested work plan (up to 100%). Or are you also interested if they could follow the precise amount of exercise? That is, more than the recommended is bad, and so is less than the recommended amount?

If you do not consider the “extra” walking/exercise, in the interventions (capping to 100%), I think some variables may yield significant differences, or it may be clearer to the reader. However, if you want to consider the “extra” walking/exercise time, then, the way is carried out at the moment is fine. This is not compulsory. It is just a suggestion for this paper or the next one. It depends on what you want to show. I guess this as the reason I was confused in the previous version.

Authors: We do understand the concern of the reviewer. Already in out previous systematic review (Albergoni et al 2019) we advocate for more clarity in defining program adherence. We know that two definitions of adherence can/will determine whether a program is considered successful or not. We did design PACE and its program adherence, with two specific aims in mind, i.  to prize participants who were active and ii. not to over-prize over-activity. Yet, in order to avoid discouraging participants we did not ignore their activity during unplanned days, even though we did distinguish it from activity conducted during planned days. Thus, we would like to stick to our calculation.         

MINOR CONCERNS

  • In Table 1, please be specific about the note at the bottom: “Faster and Slower are split according to the mean split”, particularly the mean split of what variable. Also, I would explain that the variable used for splitting the group relates to results found later in the manuscript.

Authors: This is a very good point. We have now specified where the split comes from and pointed to the right place in the text where it is explained.

  • “Expect” should be “except” I guess (page 4, line 161)

Authors: yes, typo, sorry! Rectified.

  • On page 5, the new text reads: “PACE gave credit to the participants who were also active on non-planned days.” What do you mean by giving credit? Recognition? In what way? You menioned in your answer that it was though the opaque check marks. Please include this explanation. As is, it is unclear.

Authors: We meant that PACE did acknowledge non-planned days when the target was reached with a check mark but indeed this was opaque to differentiate it from the one of planned days. We now explain this also in the methods as for comment above. We have also rephrase this sentence to improve its clarity.